# Mode-conditioning unlocks superior test-time compute scaling

**Chen Henry Wu, Sachin Goyal, Aditi Raghunathan**
Carnegie Mellon University
{chenwu2,sachingo,aditirag}@cs.cmu.edu

## Abstract

Parallel sampling is essential to test-time scaling and reinforcement learning (RL), but its effectiveness is sharply limited by diversity collapse, where models concentrate on a few modes and repeated samples produce the same mistakes. We propose the *mode-conditioning (ModC) framework*, which explicitly allocates sampling compute across reasoning modes using either specialist models or mode-specific prefixes. With predefined mode labels, ModC consistently improves test-time scaling (Pass@$k$) across controlled graph-search tasks and math reasoning benchmarks, spanning model families and sizes from 0.5B to 7B. On OpenThoughts, fine-tuning Qwen2.5-7B with ModC achieves an *4× efficiency gain* over standard training while also improving the maximum attainable Pass@$k$. We further show that gradient clustering enables ModC without predefined mode labels, yielding up to $10\%$ gains on datasets such as NuminaMath. Finally, we show that ModC improves Pass@$k$ after RL training and can further boost the Pass@$k$ gains of diversity-inducing RL methods. These results demonstrate that standard training underutilizes the diversity in data, and that ModC provides a simple, effective remedy for unlocking the full benefits of diversity in parallel sampling.

## 1 Introduction

Parallel sampling is essential to test-time scaling and reinforcement learning (RL), driving major advances in capability (Wang et al., 2023; Snell et al., 2025; DeepSeek-AI et al., 2025). In parallel scaling, the model is given multiple independent attempts, and it has proven especially effective in domains like mathematics, coding, and scientific discovery, where candidate solutions can be verified automatically, making it a backbone of systems such as AlphaEvolve (2025).

Despite its promise, parallel scaling relies on a crucial assumption: the model must generate diverse and creative solutions. In practice, however, finetuning (Dang et al., 2025) and reinforcement learning (Yue et al., 2025) are well-documented to induce *diversity collapse*, where models tend to produce the same mistakes across trials. As a result, additional samples often reproduce the same errors or converge on indistinguishable strategies, leading to diminishing returns as compute is scaled. While recent works propose ways to mitigate diversity collapse, such as Pass@$k$ training (Chen et al., 2025b) and weight-space regularization (Dang et al., 2025), we argue that a key challenge lies in using token-by-token temperature sampling for diversity, which leads to a lack of diversity in *high-level reasoning modes*.

In this work, we put forward **mode-conditioning (ModC)**, a new paradigm that explicitly structures test-time scaling around multiple reasoning modes. Consider a simple well-defined task of Countdown from Gandhi et al. (2024) which involves a search problem that admits two clear modes: breadth-first search (BFS) and depth-first search (DFS). Depending on the graph structure, both modes have their own strengths. However, as we will show, temperature sampling can fail in covering both modes for a given problem. Rather than drawing repeatedly from a collapsed distribution, we enforce coverage across strategies by conditioning on modes and allocating samples to cover diverse modes. This simple yet powerful – and, to the best of our knowledge, previously unexplored – idea provides a principled way to boost test-time scaling. The advantage becomes especially pronounced on inputs where the dominant mode fails but a lower-probability one succeeds.

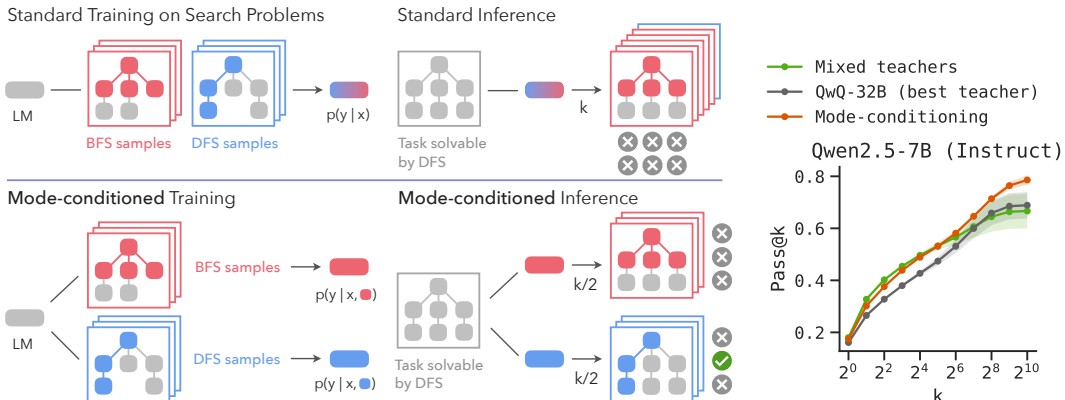

Figure 1: **Mode-conditioning for test-time scaling.** Modern LLMs often collapse to a single strategy, making Pass@$k$ scaling suboptimal: if the chosen strategy is wrong, every attempt fails. (Left) In a controlled graph task solvable by DFS or BFS, models trained on both still often commit to just one. To address this, we introduce *mode-conditioning* (ModC) that explicitly allocates test-time compute across modes. We study two training methods that enable this: separate models or a single model with mode-specific prefixes. (Right) **4× efficiency gains with ModC training.** We apply ModC to long chain-of-thought reasoning distillation on the OpenThoughts dataset. With ModC, the model achieves the same Pass@1024 as standard training using only $k = 256$ samples, yielding an ∼4× improvement in inference efficiency. Moreover, ModC also *improves* the maximum attainable Pass@$k$, pushing the frontier of test-time scaling.

While the principle of ModC is appealing, the key question is how to implement it in practice. One option is to prompt the model with explicit instructions for which mode to use, but this requires extensive manual effort to characterize modes and, more critically, models often fail to follow instructions for low-probability strategies. To move beyond such ad-hoc prompting, we train models to provide explicit control over modes. We explore two natural instantiations: (i) training *separate specialist models*, each dedicated to a distinct mode, and (ii) training a single model with *mode-specific prefixes*, where modes can be sampled reliably by using the corresponding prefix. Although both approaches are conceptually simple and have appeared in other contexts, we find that even these straightforward implementations yield substantial gains for test-time scaling.

On the task of `Countdown`, we see that standard training does in fact struggle to sample from both modes (BSF and DFS) for several inputs. With predefined mode labels, ModC with both separate models and prefixes shows **consistent superior test-time scaling** to standard training, with gains especially pronounced on those instances that can only be solved by either BFS or DFS. In this `Countdown` task, we observe that ModC with separate models outperforms ModC with prefixes, suggesting that knowledge transfer across modes is less crucial.

We apply ModC to real-world LLM reasoning datasets with both short- and long-form CoT. Our experiments fine-tune two different base LLMs, Qwen2.5-Base and OLMo2-Base across model scales from 0.5B to 7B, using distillation from two distinct teachers, each representing a different mode. Across all settings, we observe a clear and *consistent* trend: ModC significantly improves test-time scaling compared to both standard mixed-teacher training and the best single-teacher baseline. In particular, on Qwen2.5-7B, ModC training yields up to 4× efficiency gains on both long-CoT (Figure 1) and short-CoT (Figure 4). Notably, standard training often fails to exploit teacher diversity, with single-teacher models outperforming those trained on mixtures – a counterintuitive outcome, since more diverse data should in principle help. ModC, by contrast, effectively harnesses this diversity, translating it into stronger test-time scaling. In other words, diverse training data is most useful when paired with mechanisms that preserve and control its modes.

Next, we turn to ask: can ModC better harness the implicit diversity of existing post-training datasets (such as NuminaMath; Li et al., 2024) that standard training might be missing? However, applying mode conditioning requires knowing the modes apriori as well as mode annotations on the training data. We find that gradient clustering (inspired by Xia et al. (2024); Jung et al. (2025)) enables ModC

without predefined mode labels. Once again, we see consistent gains with ModC achieving up to 10% gains on the NuminaMath dataset with no additional side information.

We show that ModC improves Pass@$k$ after RL training and can further boost the Pass@$k$ gains of diversity-inducing RL methods. While RL brings both models to the same Pass@1, ModC excels immediately at $k = 2$ with higher Pass@$k$ scores. This result shows that unlike standard SFT, ModC successfully enriches the solution space without degrading the accuracy of its top output. We see similar benefits of ModC on top of Pass@$k$ RL (Chen et al., 2025b), which explicitly prevents diversity collapse during RL. This suggests that interventions designed to prevent diversity collapse during RL can be further boosted by SFT interventions such as our ModC.

These results demonstrate that standard training underutilizes the diversity in data, and that ModC provides a simple, effective remedy for unlocking the full benefits of diversity in parallel sampling. In summary, we make the following contributions.

1. **Introducing the mode-conditioning (ModC) framework.** We propose a simple but powerful paradigm to address diversity collapse in LLM reasoning and improve test-time scaling (Section 2). ModC explicitly allocates test-time compute across reasoning modes. We propose two training methods to allow for such test-time allocation: (i) specialist models and (ii) mode-specific prefixes.

2. **ModC demonstrates consistent gains across tasks.** Through controlled graph-search experiments (Section 3) and large-scale reasoning benchmarks (short- and long-form CoT, distillation from multiple teachers) (Section 4), we show that ModC achieves substantial and consistent improvements in test-time scaling, including up to $4\times$ efficiency gains. We also carefully analyze tradeoffs between different ModC training methods, effect of model size, data composition etc.

3. **ModC on training data without predefined mode labels.** We find that gradient clustering enables ModC without predefined mode labels, yielding up to 10% gains on datasets such as NuminaMath. These results suggest that standard training underutilizes the diversity in data—an inefficiency that the ModC framework directly addresses.

4. **The benefit of ModC holds on top of standard RL and diversity-inducing RL.** Unlike standard SFT, ModC successfully enriches the solution space and mitigates diversity collapse during RL. Results on top of Pass@$k$ RL suggest that interventions designed to prevent diversity collapse during RL can be further boosted by SFT interventions such as our ModC.

## 2 THE MODE-CONDITIONING FRAMEWORK

### 2.1 PRELIMINARIES

Large language models (LLMs) generate outputs by sampling from a probability distribution over continuations. In more complex tasks, instead of producing a single output, we can allocate additional *test-time compute* by drawing $k$ independent samples for the same input and selecting the best candidate. This strategy, known as *parallel scaling*, is especially effective in tasks where solutions can be automatically verified (e.g., mathematics or programming). The performance of this standard approach is captured by the Pass@$k_{\texttt{std}}$ metric on an input $x$:

$$\text{Pass@}k_{\texttt{std}}(x) = 1 - (1 - p_x)^k, \tag{1}$$

where $p_x$ is the Pass@1 or probability of sampling a trajectory that is successful on input $x$.

In practice the gains with parallel scaling depend strongly on the underlying success probability $p_x$. Modern training pipelines such as supervised fine-tuning and reinforcement learning often induce *mode collapse*, where the model commits to a small set of strategies (Dang et al., 2025; Yue et al., 2025; Sessa et al., 2024; Chow et al., 2024). On some prompts, this collapse drives the probability $p_x$ of sampling a successful strategy to be very small, so that an impractically large number of samples is required to obtain good performance.

### 2.2 MODE-CONDITIONED TEST-TIME SCALING

One approach is to modify the finetuning objective to prevent collapse and maintain higher $p_x$. We take a complementary route: rather than sampling from a single collapsed distribution, we explicitly allocate test-time compute across *diverse modes*, enforcing coverage so samples include not only the dominant strategy but also alternatives that may succeed where it fails.

Consider two modes with success probabilities $p_{1,x}$ and $p_{2,x}$. If we split the budget evenly, sampling $k/2$ trajectories from each mode, the resulting probability of solving input $x$ is

$$\text{Pass@}k_{\text{ModC}}(x) \;=\; 1 - (1 - p_{1,x})^{k/2}(1 - p_{2,x})^{k/2}. \tag{2}$$

In contrast, suppose the model does *not* know *a priori* which mode is better for $x$. Let $w_x \in [0,1]$ denote the (random) probability with which the model uses the first mode, and assume that the model is unbiased between the two modes, in the sense that $\mathbb{E}[w_x] = 1/2$ (e.g., $w_x$ is drawn from any distribution centered at 0.5). A single sample from the model succeeds with probability $w_x p_{1,x} + (1 - w_x)p_{2,x}$, so we have

$$\text{Pass@}k_{\text{std}}(x; w_x) = 1 - (1 - w_x p_{1,x} - (1 - w_x)p_{2,x})^k. \tag{3}$$

The function $w \mapsto 1 - (1 - p_{2,x} - (p_{1,x} - p_{2,x})w)^k$ is concave on $[0,1]$, so by Jensen's inequality

$$\mathbb{E}_{w_x}\big[\text{Pass@}k_{\text{std}}(x; w_x)\big] \;\leq\; \text{Pass@}k_{\text{std}}\big(x; \mathbb{E}[w_x]\big) = \text{Pass@}k_{\text{std}}(x; 1/2). \tag{4}$$

When $w_x = 1/2$, the single-sample success rate is $p_x = (p_{1,x} + p_{2,x})/2$. Whenever $p_{1,x} \neq p_{2,x}$, we have

$$(1 - p_{1,x})^{k/2}(1 - p_{2,x})^{k/2} < (1 - p_x)^k, \tag{5}$$

which implies

$$\text{Pass@}k_{\text{ModC}}(x) > \text{Pass@}k_{\text{std}}(x; 1/2) \;\geq\; \mathbb{E}_{w_x}\big[\text{Pass@}k_{\text{std}}(x; w_x)\big]. \tag{6}$$

In other words, even when the model's mode-usage follows *any* distribution centered at $0.5$, explicitly allocating compute evenly across modes is strictly better than sampling from the model's own uncertain mixture.

But how do we explicitly sample from different modes in practice? A naive baseline is to prompt the model with instructions to use different strategies. However, this approach is unreliable: it is unclear how to phrase the prompts, and the model may not consistently follow them.

## 2.3 MODE-CONDITIONED TRAINING

Instead of relying on ad-hoc prompting to elicit different behaviors, we consider scalable training objectives that explicitly enforce control over modes. This provides a reliable lever at test time for allocating compute across diverse strategies. We explore two natural instantiations: training with separate specialist models and training with prefixes within a single model.

In this section, we assume that the relevant modes are known *a priori* and that training data can be annotated with the mode used to generate each trajectory. While this assumption is convenient for exposition and testing the benefits of our paradigm, it is not strictly necessary. In practice, one could imagine automated approaches for mode discovery and annotation, for example by clustering trajectories using gradient-based similarity measures or other unsupervised techniques. We return to this point in §5 where we discuss how mode-conditioned training can be extended to settings where the modes are not explicitly labeled in the data.

**Mode-conditioned training with separate models.** The most direct approach is to train distinct models, each specialized to a particular mode of reasoning. Concretely, the training data is partitioned into subsets corresponding to different strategies, and a separate model is trained on each subset while keeping total training data and compute constant. At test time, the sampling budget is divided across the specialists (e.g., $k/2$ samples from each in the two-mode case). This design ensures strong specialization and reduces correlated errors, which translates into more effective parallel scaling.

**Mode-conditioned training with prefixes.** While separate models improve diversity, they prevent knowledge sharing across modes. This is a significant drawback in realistic reasoning tasks, where different strategies often rely on common linguistic or mathematical foundations.

To overcome this, we draw inspiration from the literature on steering model behavior via *explicit conditioning tokens*, a technique used widely in controlled text generation (Keskar et al., 2019) and instruction tuning. We prepend discrete condition tokens (e.g., [Mode 1], [Mode 2]) to the input,

training the model to associate each prefix with a distinct reasoning strategy. At inference, balanced compute allocation is enforced by sampling evenly across the conditioning prefixes. This allows the model to specialize into distinct modes while still sharing knowledge across them, making it more scalable than training separate specialist models.

# 3 INVESTIGATING MODE COVERAGE IN PARALLEL SAMPLING

## 3.1 THE Countdown TASK

Countdown is a generalization of *Game of 24*, where a model must find a sequence of arithmetic operations to transform a set of starting numbers into a target value (Gandhi et al., 2024). Given several starting numbers, the model can apply operations $\{+, -, \times, \div\}$ to reach a target. For example, given $\{10, 10, 4, 6\}$ with target 16, one solution is $(10 \times 10 - 4) \div 6 = 16$.

This task naturally admits two distinct problem-solving modes: depth-first search (DFS) and breadth-first search (BFS). This allows us to precisely control and examine which mode is used. Solutions are easily verifiable by checking if operations reach the target, which makes it an ideal testbed for studying test-time scaling with parallel sampling.

## 3.2 WHY MODE COVERAGE IS IMPORTANT?

In principle, both BFS and DFS are complete search algorithms capable of finding any solution, so why would mode coverage matter for this task? For real-world problems, however, computational constraints require using heuristics to make the search tractable. Following Gandhi et al. (2024), we use heuristics to guide and prune the search space, which introduces an important asymmetry: with heuristic pruning and search budget constraints, each algorithm excels on different problem instances – some problems become solvable only by the oracle DFS while others only by the oracle BFS. Since we cannot know *a priori* which algorithm will succeed for a given problem, maintaining coverage of both modes during test-time sampling becomes crucial for achieving high success rates.

To evaluate test-time scaling, we report Pass@$k$ metrics on two held-out test sets. We first create a ***natural*** test set of 500 problems by randomly selecting unseen target numbers and valid starting numbers that can reach the target. Second, we subsample an ***adversarial*** test set designed to require mode diversity: we filter for problems where either oracle BFS or oracle DFS (but not both) achieves less than $5\%$ success rate across multiple runs. This adversarial set directly tests mode coverage – each problem is effectively solvable by only one algorithm, so achieving high accuracy requires sampling from both DFS and BFS modes when we do not know *a priori* which one works better. We expect that the benefit from explicit test-time balancing is much larger on the adversarial test set.

## 3.3 MODE COVERAGE

We use rejection sampling to create our training set, keeping only instances where at least one search algorithm (oracle DFS or BFS) successfully finds a solution. Specifically, we uniformly generate a target number from 1 to 200 and four starting numbers that can reach the target, and uniformly choose DFS and BFS and one of the search heuristics for guidance and pruning. We note that DFS has a higher overall success rate, so our final training set ends up consisting of 163K problems, with 97K DFS solutions and 65K BFS solutions. Each training example includes the input, the search trajectory, and final operations. We train Qwen2.5-Base models (0.5B–7B) for 4 epochs.

We first examine whether models trained on mixed DFS and BFS data can learn to balance the two algorithms under repeated sampling. Figure 2 shows the fraction of BFS used by the model for each test problem. We observe that standard training on the mixture of both algorithms (shown in gray) tends to bias toward one algorithm for many test problems, i.e., either predominantly using DFS (low BFS fraction) or BFS (high BFS fraction), rather than balancing both.

**Effect of diversity of training data.** Recall that we use rejection sampling to create our training set, which naturally biases the training data towards the on-average more promising algorithm (i.e., DFS in this case). What if we had 50-50 data for DFS and BFS? To answer this, in Figure 2 we also plot the distribution for this standard training with balanced data. We see that the distribution is less skewed, but still many problems have extremely imbalanced allocation of test-time compute.

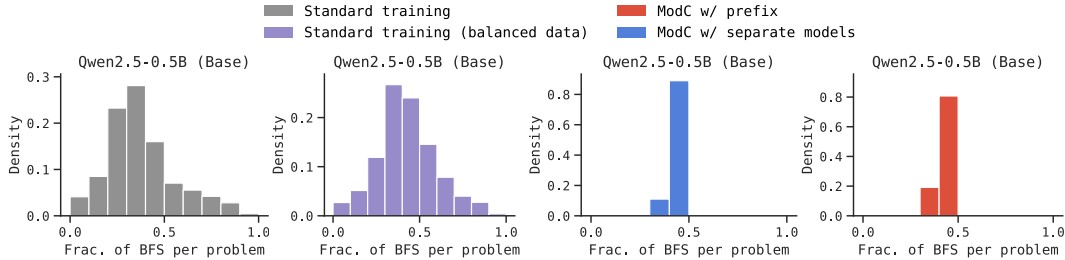

Figure 2: **Standard training fails to balance diverse modes *per problem* under repeated sampling.** This issue does not go away with balanced training data. Instead, ModC explicitly targets and successfully achieves balanced test-time compute allocation.

**ModC balances test-time compute allocation.** In contrast, ModC with explicit balanced allocation achieves the desired behavior. We see that for both training separate models (shown in blue) and training with prefixes (shown in red), the fraction of BFS per problem is concentrated around 0.5, which demonstrate nearly perfect balance.

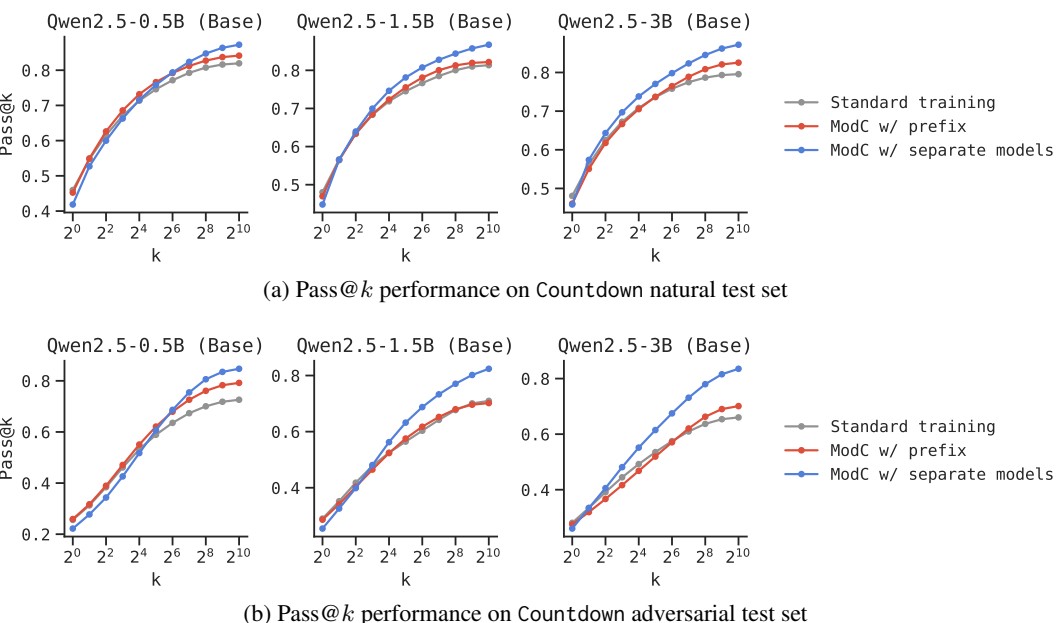

(a) Pass@$k$ performance on `Countdown` natural test set

(b) Pass@$k$ performance on `Countdown` adversarial test set

Figure 3: **Balanced test-time allocation improves scaling.** (a) On the natural test set of `Countdown`, balanced test-time allocation with ModC shows consistent improvements as $k$ increases. (b) On the adversarial test set where each problem is challenging for one one algorithm (oracle DFS or BFS), the gains from enforced mode diversity are even more pronounced.

### 3.4 FROM MODE COVERAGE TO PASS@$k$

In this part, we show that ModC dramatically improves test-time scaling, particularly on problems that require diverse algorithmic modes. We start with ModC with separate models. Figure 3 shows the results on the natural and adversarial test set of `Countdown` across model scales. While Pass@1 is comparable or slightly lower for separate models, the scaling behavior dramatically improves by up to 8% for Pass@1024. This gap is consistent across all model sizes, with separate models maintaining superior scaling curves. The advantages are even more pronounced on the adversarial test set, where we see that the gap boosts up to 20% for Pass@1024. This shows that mode balance is indeed crucial for test-time scaling – problems that DFS finds hard to solve are potentially solvable for BFS, which enable better coverage of the problem space. On the other hand, Figure 3 shows that ModC with prefix with balanced allocation outperforms standard training for most scales. As a control, we try

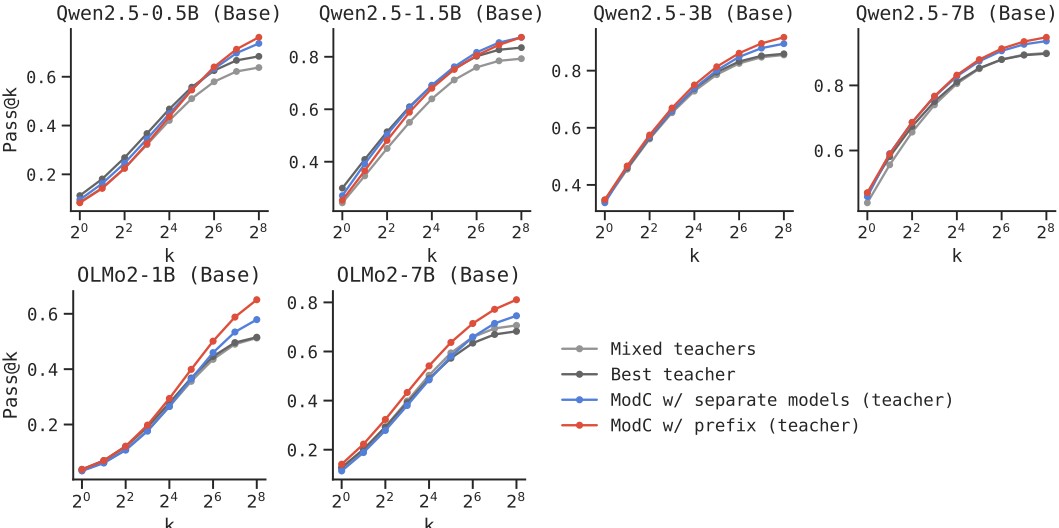

Figure 4: **ModC improves short CoT reasoning.** Pass@$k$ on MATH500. Naively mixing teacher data underperforms the single-teacher baseline, while ModC shows consistent gains. ModC with prefixes generally works better than ModC with separate models underscoring the importance of sharing knowledge across modes (teacher strategies) in math reasoning.

random partitioning the training data into two groups, which sometimes shows gains but does not outperform ModC (see details in §B).

## 4   MODE-CONDITIONING IMPROVES MATH POST-TRAINING

We saw how ModC improves Pass@$k$ performance on `Countdown` and is superior for parallel scaling. In this section, we evaluate ModC when post-training for math reasoning. We investigate both long and short CoT reasoning across multiple model families and scales.

### 4.1   DISTILLATION FROM MULTIPLE TEACHERS

We start with a natural source of diverse modes: distillation from multiple stronger teacher models. This setting is particularly relevant given the existence of strong models with distinct reasoning and response styles. Recent work typically selects the single teacher that provides the best distillation performance (Muennighoff et al., 2025; Guha et al., 2025). We test whether explicitly balancing compute across multiple teacher strategies improves performance. In each setting, we collect CoT reasoning traces from two teacher models for mathematical problems, where each teacher demonstrates distinct problem-solving styles. Following the methods established in Section 3, we compare four training strategies: (1) *mixed teachers*: mixing all teacher data together, (2) *best teacher*: training on the teacher data that produce the best performance (3) *ModC with separate models*: training separate models on each teacher's data independently, and (3) *ModC with prefixes*: prepending teacher identity tokens (e.g., using `[teacher]-style reasoning`) to each CoT and using balanced test-time allocation where we sample equally from each teacher condition.

### 4.2   SHORT CHAIN-OF-THOUGHTS

**Experimental setup.**   We first experiment with post-training NuminaMath (Li et al., 2024) dataset where the chain-of-thought completions are relatively short and do no involve long thinking. We use the SFT traces distilled from two teacher models: DeepSeek-R1 (DeepSeek-AI et al., 2025) and GPT-OSS-120B (OpenAI, 2025), with the problems from NuminaMath. For evaluation, we use MATH500 (Hendrycks et al., 2021) and measure Pass@$k$. We post-train Qwen2.5-Base (0.5B–7B) and OLMo2-Base (1B–7B) models for 4 epochs. We tune the learning rate $\in \{1e\text{-}4, 1e\text{-}5\}$ and use AdamW optimizer with global batch size of 256.

**Naive data mixture underperforms.** Figure 4 compares the Pass@$k$ curves on MATH500 for the four training strategies mentioned above. We observe that naively mixing data from both teachers either underperforms or is at best comparable to the stronger single-teacher baseline. This is consistent with prior intuition that training on the best teacher can be more effective than mixing teachers.

**ModC training unlocks superior test-time scaling.** On the other hand, training on both teachers' data but with mode conditioned (ModC) training and inference unlocks better test-time scaling (Figure 4). The gains are consistent across model families (Qwen and OLMo2) and scales (0.5B to 7B), offering up to 10% gain on Qwen2-0.5B and 15% on OLMo-2-7B. Comparing the two variants of ModC, we see that ModC with prefixes generally outperforms ModC with separate models suggesting that knowledge sharing across modes is crucial for math tasks.

### 4.3 LONG CHAIN-OF-THOUGHTS

We now examine long CoT reasoning, where models usually spend tens of thousands of tokens on extended reasoning before producing the answer on more challenging problems.

**Experimental setup.** We use the subset of problems from OpenThoughts-3 (Guha et al., 2025) that they did ablation studies for the teacher models with. Specifically, solutions are from two teachers: QwQ-32B (Team, 2025) and DeepSeek-R1 (DeepSeek-AI et al., 2025). For evaluation, we use AIME 2025 and measure Pass@$k$. Following Guha et al. (2025), we initialize model weights with Qwen2.5-7B-Instruct.

**4× efficiency gains with ModC.** Figure 5 shows similar patterns to short CoT: ModC achieves consistently higher Pass@$k$ than both single-teacher and mixed-teacher baselines. Even in the long CoT setting with extremely large token budgets per sample, standard training fails to adequately cover multiple modes—mixed training again does not outperform the best single teacher. In contrast, ModC not only provides an effective way to learn from multiple teachers and surpass each individual teacher, but also delivers substantial efficiency gains: it matches the Pass@1024 of standard training with only $k = 256$ samples, yielding nearly **4× faster inference**, while simultaneously pushing the maximum achievable Pass@$k$. We

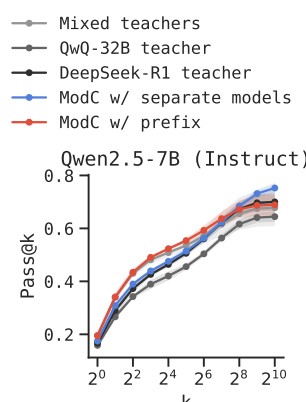

Figure 5: **ModC improves long CoT reasoning.** Pass@$k$ on AIME 2025. Standard training with multiple teachers fails to outperform the best single teacher, but ModC with multiple teachers surpasses single teacher.

### 5 AUTOMATED MODE DISCOVERY

We have seen that ModC improves test-time scaling across a wide variety of real-world settings. However, previous sections assume access to natural modes in the data: search algorithms in `Countdown` or teacher identities in multi-teacher distillation. In practice, most real-world training data lacks such clear segregation. Can we extend ModC to work on training data that contains mixed modes but lacks explicit labels? Furthermore, can we discover meaningful modes in training data without apriori knowledge of these modes? We explore both questions in this section, applying gradient clustering to discover and annotate modes in training data.

### 5.1 GRADIENT CLUSTERING

Gradient similarity has been shown effective in understanding training dynamics (Jacot et al., 2018), identifying influential training data (Koh & Liang, 2017), and diversifying data selection (Jung et al., 2025). That inspires us to test whether gradient cluster can discover meaningful modes in training data that we should condition on.

For each training example $(x, y)$, we compute gradients with respect to model parameters

$$g_\theta(x, y) = \nabla_\theta \log p_\theta(y|x).$$

To reduce dimensionality, we follow prior works (Xia et al., 2024; Jung et al., 2025) to apply Rademacher random projection (Park et al., 2023) to each gradient vector. Once we get the projected gradient vectors for all training samples, we cluster the vectors into $C$ clusters, and all samples in the same cluster belong to one "mode" based on which we apply ModC.

## 5.2 Gradient clustering improves post-training on general data

We apply gradient clustering to NuminaMath, a real-world dataset that is probably quite diverse but we lack a clear sense of what modes exist. Surprisingly, we see that training with ModC on automatedly discovered modes (via gradient clustering) yields significant improvements. Figure 6 shows that ModC consistenty improves Pass@$k$ compared to standard training across model scales.

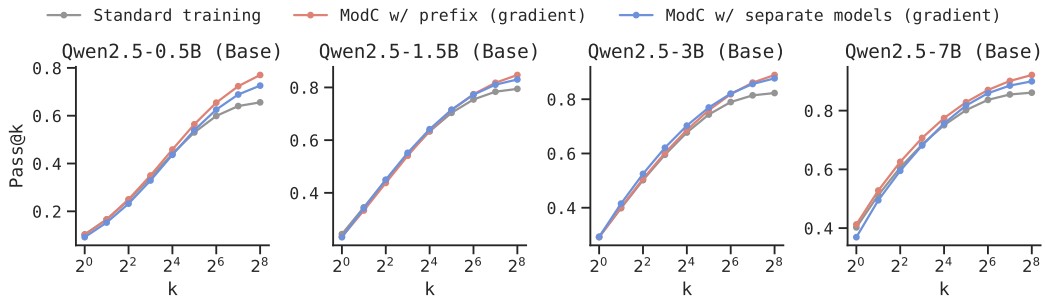

Figure 6: **ModC on automatedly discovered modes via gradient clustering improves short CoT.** Pass@$k$ on MATH500 for Qwen2.5-Base models finetuned with standard finetuning and ModC.

## 6 Mode-Conditioning Improves Reinforcement Learning

Finally, we turn to verify if the benefit of ModC holds after RL, where the distribution becomes sharper and diversity decreases. For RL methods: we consider (1) standard RL, where the distribution becomes sharper and diversity decreases, and (2) Pass@$k$ RL (Chen et al., 2025b), which can improve both Pass@1 and Pass@$k$. For each RL method, we initialize the policy from either the standard SFT models or ModC models. For the prefix variant, we believe we need prefix-following rewards to make sure that the learned prefix-mode binding is maintained. Therefore, we focus on the separate models variant given its simplicity and leave RL on the prefix variant for future work.

Figure 7 shows the results of ModC vs standard SFT followed by different RL methods. Crucially, while RL brings both models to the same Pass@1, ModC excels immediately at $k = 2$ with higher Pass@$k$ scores. This result shows that unlike standard SFT, ModC successfully enriches the solution space without degrading the accuracy of its top output. Moreover, we see that this observation also

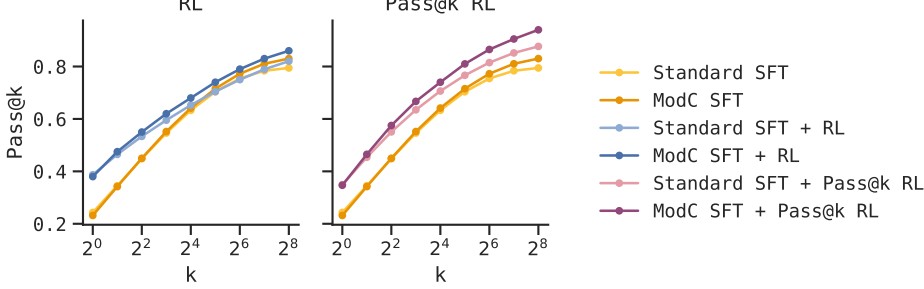

Figure 7: **ModC improves long CoT reasoning.** Pass@$k$ on MATH500. While RL brings both models to the same Pass@1, ModC excels immediately at $k = 2$ with higher Pass@$k$ scores.

holds for Pass@$k$ RL, which explicitly prevents diversity colllapse during RL training. This suggests that interventions designed to prevent diversity collapse during RL can be further boosted by SFT interventions such as our ModC.

## 7 RELATED WORK

**Improving parallel test-time scaling.** Repeated sampling significantly improves the performance of LLMs especially in domains like reasoning and coding (Wang et al., 2023; Brown et al., 2024; Rozière et al., 2024). To decide the final answer from the $k$ attempts, one can use either majority voting (Wang et al., 2023) or a verifier (especially in the code and scientific discovery domains) (Wang et al., 2024; AlphaEvolve, 2025). However, a series of works (Cobbe et al., 2021; Dang et al., 2025) have identified issues during post-training of LLMs that impair the diversity in model generation, consequently affecting the efficacy of test-time scaling. Huang et al. (2024) attribute this to the sharpening effect whereas Chu et al. (2025) highlight memorization as a root cause.

Lot of recent works have in turn have proposed fixes for improved parallel test-time scaling. Beeching et al.; Snell et al. (2024) propose modifications to beam search to explciitly optimize for diversity amongst the candidates. Wang et al. (2025); Hughes et al. (2024) propose diverse prompting to improve test-time scaling. Taking a step back, Sessa et al. (2024); Chow et al. (2024); Chen et al. (2025a) explicitly optimize for best-of-k performance during the finetuning process. Dang et al. (2025) propose a simple fix of ensembling the finetuned weights with the base model to mitigate diversity collapse. Goyal et al. (2025) further take a step back and propose pretraining with logit distillation to improve parallel test-time scaling behaviors. In contrast, in this work we propose a more data centric conditioning of the finetuning process to explcitly encode specialist modes in the model.

**Creativity and diversity of language models.** Recent work has evaluated creativity and diversity in language models with mixed findings. Models underperform human experts in creative writing (Chakrabarty et al., 2024) and humor generation (Zhang et al., 2024). While Anderson et al. (2024) find that language models increase individual idea divergence but causes group-level homogenization, Si et al. (2024) report that LLMs can generate novel research ideas despite feasibility limitations. Nagarajan et al. (2025) demonstrate that global planning and seed-conditioning are crucial for creative generation. Temperature sampling weakly correlates with creativity and often introduces incoherence (Chen & Ding, 2023). On the evaluation front, various benchmarks have been proposed to measure output diversity and creativity (McLaughlin et al., 2024; Zhang et al., 2025; Jansen et al., 2024).

**Specialization in model training.** In this work, we proposed *ModC* where a single model is explicitly conditioned to learn separate modes of finetuning data. We do this by prepending conditioning tokens (e.g., DFS or BFS) to the reasoning traces. Closest to our work is Mixture-of-Experts (Shazeer et al., 2017; Jiang et al., 2024; Jelassi et al., 2025) where different datapoints are routed to a specific subpart of the model which is expert for the domain of the particular datapoint. However, in contrast in ModC all the datapoints are processed by the whole model and not a subpart. Mixture-of-experts are aimed at reducing the active parameter footprint of the model.

## 8 CONCLUSIONS AND FUTURE WORK

In this work, we demonstrated that deliberate mode-conditioning (ModC) during training and inference is crucial for unlocking the full benefits of test-time compute scaling. Across both controlled search problems and large-scale reasoning benchmarks, we showed that standard training tends to collapse onto a single strategy, while specialization through mode-conditioning consistently yields superior scaling and more reliable gains. Beyond explicit labels such as teacher identity, we further showed that gradient clustering can automatically uncover meaningful specializations, making ModC broadly applicable. We finally show that ModC mitigates diversity collapse during both standard RL and diversity-inducing RL. It suggests that interventions designed to prevent diversity collapse during RL can be further boosted by SFT interventions such as our ModC. Looking ahead, exciting directions include extending ModC to a larger number of modes and to richer behavioral dimensions such as reasoning depth or planning style, as well as integrating ModC with adaptive allocation policies that learn how to optimally divide compute across modes at test time. Another promising avenue is further improving ModC with reinforcement learning, where balanced modes could encourage diverse exploration early in training and be gradually relaxed. Together, these directions position mode-conditioning as a general and effective principle for building more powerful reasoning systems.

## 9    REPRODUCIBILITY STATEMENT

We have taken several steps to ensure the reproducibility of our results. Complete experimental details – including model architectures, optimization hyperparameters, datasets, and evaluation protocols – are provided in the relevant sections of the main paper for each experiment. In addition, we submit our code and scripts as part of the supplementary material, and will be open-sourcing it along with other meta-data upon acceptance of the work.

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

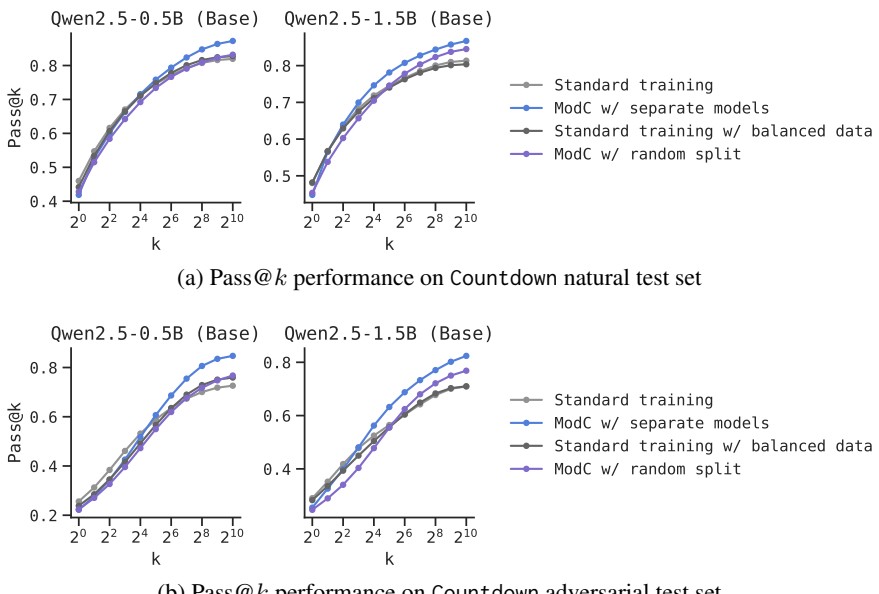

(a) Pass@$k$ performance on `Countdown` natural test set

(b) Pass@$k$ performance on `Countdown` adversarial test set

Figure 8: **Ablation studies on `Countdown`.** ModC with random paritioning sometimes shows gains but does not outperform ModC with DFS/BFS partition. Balanced training data DFS/BFS distribution does not show gains compared to standard training.

## A    LLM USAGE

We used a large language model (LLM) solely to refine the writing style. The LLM played no role in research ideation, experimental design, or analysis. All technical content, results, and conclusions are entirely our own, and we take full responsibility for the final manuscript.

## B    ADDITIONAL RESULTS FOR `Countdown`

We see that ModC with DFS/BFS partition improves test-time scaling on `Countdown`. As a control, we also try random partitioning the training data into the same number of groups. From Figure 8, we see that ModC with random paritioning sometimes shows gains but does not outperform ModC with DFS/BFS partition. Another baseline we try is to enforce 50-50 distribution of DFS and BFS in the training data, which we do not see any gains compared to standard training.

## C    ADDITIONAL RESULTS FOR GRADIENT CLUSTERING

**Gradient clustering recovers teacher identity.**    We first validate gradient clustering on multi-teacher data where ground-truth labels exist. Using the short CoT dataset from Section 4.2, we compute gradients using Qwen2.5-Base 1.5B and apply clustering. We first observe that gradient clustering achieves $98.7\%$ F1 score in recovering teacher assignments. More importantly, Figure 9 shows that ModC with these automatedly discovered gradient-based clusters yields nearly identical test-time scaling benefits as using true teacher labels. This confirms that gradient patterns effectively capture the underlying modes.

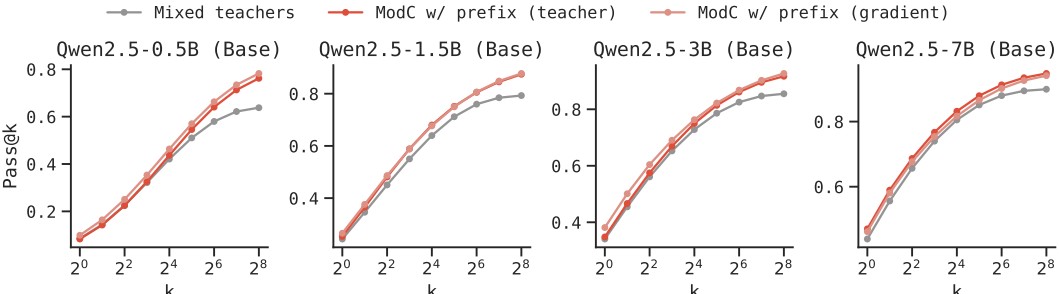

Figure 9: **Validating gradient clustering on multi-teacher data.** ModC on training data that is distilled from multiple teachers outperforms standard training even without access to teacher identity annotation on training data. ModC with gradient clustering almost completely matches ModC with access to teacher annotations.

