# OpenReview forum: "Mode-conditioning unlocks superior test-time compute scaling"
_ICLR.cc/2026/Conference — ICLR 2026 Poster_

### Official Review · Reviewer_myXD · 2025-11-01

**Soundness:** 3
**Presentation:** 4
**Contribution:** 3
**Rating:** 8
**Confidence:** 3

**Summary:**

This work suggests a method called ModC for improving diversity in generation during parallel scaling.

The idea is that different ways to approach a problem may fall into different "modes", corresponding to different broad strategies. A diverse generator of possible proofs should try to sample as diversely as possible from different modes.

This paper suggests two possible ways to do this: (1) train a separate model on each mode, or (2) train a model with prefix-tuning to make it use one mode.

Modes can be either (1) known a priori, or (2) found automatically with a gradient clustering method. The paper tests the idea on several benchmarks such as NuminaMath, AIME, and Countdown, and finds benefits over vanilla parallel scaling -- especially when there is a large amount of parallel scalin.g

**Strengths:**

* ModC has a conceptually clear motivation.

* The proposed method is simple to implement, and seems to yield increased performance. This might be of interest to much of the ICLR  community, since methods to improve to model reasoning are quite popular.

* The paper presents a way to find modes automatically, using gradient cluster. This makes it more broadly applicable than it would be if the modes had to be created manually.

* The experimental methodology seems mostly sound (although I have a question -- see weaknesses below).

**Weaknesses:**

* On the methodology: I'm not sure how good of a metric pass@k is for AIME, when k = 1000, because there are only 1000 possible solutions for any problem as far as I know.
    - Having a model that outputs a random number from 0 to 999 would give a 63% pass@k accuracy, which is roughly the accuracy  reported in Figure 5.
    - On the other hand, having 1000 models (each of which outputs a constant number) would give a 100% pass@k accuracy.

* There's a quite relevant prior work called "Metadata Conditioning Accelerates Language Model Pre-training" by Gao et al., 2025, that this made me think about. There they show that adding metadata of which website a text came from can improve model performance. It could be good to discuss the connection with this work.

**Questions:**

Typos: "this achieves up to xx% improvement"

---

> ### Author Response · Authors · 2025-11-22
>
> Thank you for your careful and encouraging review! We address your comments and concerns below.
>
> > **For AIME, pass@k with k = 1000 may not be a very informative metric, since there are only 1000 possible solutions. A random model outputting a uniform number from 0 to 999 would achieve about 63% pass@k, and 1000 constant-output models would give 100% pass@k.**
>
> We appreciate this insightful observation. We’d like to argue that “the answer should be an integer between 1 and 1000” is very privileged information that’s unknown to the model when we test it. We are adding this potential concern and the discussion in the new version.
>
> > **There is a relevant prior work, “Metadata Conditioning Accelerates Language Model Pre-training” (Gao et al., 2025), that shows adding metadata (such as website source) can improve model performance. It would be good to discuss the connection with this work.**
>
> Thank you for pointing this out. While our main focus is on inference-time scaling, we agree that this work is relevant. We hope our work on inference-time scaling at post-training stage provides an interesting motivation for doing metadata conditioning during pretraining, e.g., a metadata-conditioned pretrained model can be better at following patterns in the mode. We are adding the above discussion in the new version.
>
> Additionally, we’d like to also refer you to our new results on demonstrating that the efficacy of ModC holds after both standard RL and pass@k RL training (see responses to Reviewers MR1M and ghYh). Specifically, we see that:
>
> 1. Crucially, while pass@k RL brings both models to the same pass@1, ModC excels immediately at k=2 with higher pass@k scores. This shows that unlike standard SFT, ModC successfully enriches the solution space without degrading the accuracy of its top output.
> 2. The finding also holds for standard RL.
>
> Let us know if you have other questions!

---

### Official Review · Reviewer_ghYh · 2025-11-02

**Soundness:** 3
**Presentation:** 2
**Contribution:** 3
**Rating:** 4
**Confidence:** 4

**Summary:**

This paper studies mode conditioning as a way to address the lack of diversity in LLM generations for reasoning tasks. The problem is that when we scale test time compute with parallel sampling, current models tend to collapse to one or two dominant strategies, so additional samples mostly repeat the same errors. The paper’s proposal is to make the modes explicit and to allocate test time samples across modes rather than drawing all samples from a single collapsed distribution. They instantiate this in a controlled search setting (Countdown, a generalization of Game of 24) where the target value can be found either by a DFS style search or a BFS style search, and where the search trace itself reveals which mode was used. They then extend the idea to math post training with multiple teacher models and finally to a setting where modes are discovered automatically via gradient clustering.  They show that mode conditioned training and mode conditioned inference improves Pass@k relative to standard mixed training.

**Strengths:**

I think the problem is interesting and well chosen. How to obtain diversity in reasoning style without simply increasing sampling temperature (which has its own issues) is still not well understood, and most current post training pipelines / RL algorithms do in fact make diversity worse.

I like the synthetic setup with countdown game since it cleanly isolates the question they are trying to answer, with a way to verify which mode of problem solving is used.  The experiments are pretty thorough and they also show some nice results in the math CoT setting, as well as in the automatic mode finding setting using gradient clustering.

**Weaknesses:**

1. Several parts of the paper are somewhat hard to follow on first read. One example is Figure 2. It is not completely clear how the per problem histograms are computed. My reading is that for each test problem the authors sample the model repeatedly, detect for each sample whether the model used DFS or BFS, compute the fraction of BFS samples for that problem, and then plot the distribution of that fraction over all problems. If that is correct, the number of samples per problem needs to be stated. If that is not correct, the figure needs a more explicit description. Right now it is difficult to tell what exactly is being compared.
2. In the separate model setting each mode gets its own model trained on the subset of data for that mode. In Countdown the paper notes there are about 97k DFS trajectories and 65k BFS trajectories. If each of those is used to train a full model of size $N$ then the total training compute for the separate model setting is roughly $6 \times (2 N) \times (97+65)/165 \approx 11.7 N D$  flops, whereas the standard training or prefix based modC is roughly 6ND .  This would mean that the separate model setting is using roughly twice the training compute.  The paper should clarify whether training budget was controlled, whether epochs were scaled down for the separate models, or whether the comparison is intentionally not compute matched. As written, it is not a fair comparison.
3. A natural application of this work is to post train with RL, where we know that the distribution becomes sharper and diversity decreases. As far as I can tell, the paper only considers SFT / distillation like settings. A natural question is: after RL, does mode conditioning still preserve the benefits shown here, or does RL erase them. It would be useful to see in the same synthetic Countdown setting a comparison of RL that samples from the usual policy versus RL that is constrained to use mode conditioned sampling during rollouts. If the authors can show that they can keep the sampling gains after RL training (evaluating 0-shot after RL) that would be a nice finding, even if on a synthetic task.
4.  There are a few grammatical and formatting issues. For example Section 5.1 appears to have an incomplete closing sentence. (Did not penalize for this.)

I am willing to improve my score if the authors can meaningfully address these comments.

**Questions:**

1. Unclear algorithm in Section 4.2 - The post training described for math reasoning in Section 4.2 seems to be plain SFT on two teacher traces with either mode specific prefixes or separate models. It would be good to state clearly that no RL was used here, if that is in fact the case.
2. Figure 4 interpretation - The caption says that the dark gray line is “best teacher.” Does this mean this curve corresponds to distillation from only the best single teacher, not distillation from the union of best teacher traces across problems?

---

> ### Author Response · Authors · 2025-11-22
>
> Thank you for your detailed review! We address your comments and concerns below.
>
> > **Several parts of the paper are somewhat hard to follow on first read. One example is Figure 2. It is not completely clear how the per problem histograms are computed. My reading is that for each test problem the authors sample the model repeatedly, detect for each sample whether the model used DFS or BFS, compute the fraction of BFS samples for that problem, and then plot the distribution of that fraction over all problems. If that is correct, the number of samples per problem needs to be stated.**
>
> We apologize for the lack of clarity and are revising the paper to clarify this. Your interpretation is correct. For each test problem we sample from the model **256** times, detect for each sample whether the model used DFS or BFS, compute the fraction of BFS samples for that problem, and then plot the distribution of that fraction over all problems. Sampling 256 times ensures that the estimation of fraction has a very low variance.
>
> > **In the separate model setting, it appears that training compute might be roughly twice that of the standard or prefix-based settings. It is unclear whether training budget was controlled; as written, the comparison does not seem compute-matched.**
>
> This is a great question, and we would like to clarify that the comparisons are compute-matched: when we train separate models, each model was trained on half the number of steps, so the total number of training steps for each method is the same.
>
> > **A natural application is RL post-training, where the distribution becomes sharper and diversity decreases. The paper currently only considers SFT/distillation-like settings. It would be useful to see whether mode conditioning still preserves benefits after RL, e.g., comparing RL that samples from the usual policy versus RL constrained to mode-conditioned sampling during rollouts.**
>
> This was a great suggestion, and we have very exciting new results over the past week. We focused on the math setting and used gradient-clustered modes. For RL methods: we consider (1) standard RL, where the distribution becomes sharper and diversity decreases, and (2) pass@k RL, which can improve both pass@1 and pass@k. We need to verify if ModC’s benefit holds after standard RL or pass@k RL. Given the time, we used 0.5B models as RL is very slow to train. We updated the results in Figure 7 of the latest version, and summarize the findings below:
>
> 1. Crucially, while pass@k RL brings both models to the same pass@1, ModC excels immediately at k=2 with higher pass@k scores. This shows that unlike standard SFT, ModC successfully enriches the solution space without degrading the accuracy of its top output.
> 2. The finding even holds for pass@k RL.
>
> We hope these experiments and results address your biggest concern!
>
> > **The post-training described for math reasoning in Section 4.2 seems to be plain SFT on two teacher traces with either mode-specific prefixes or separate models. Could you state clearly that no RL is used there, if that is in fact the case?**
>
> We added the RL results in Figure 7 of the latest version (see a summary in the above point). The original experiments in Section 4.2 are purely SFT/distillation.
>
> > **In Figure 4, the caption says that the dark gray line is “best teacher”. Does this mean this curve corresponds to distillation from only the best single teacher, not from the union of best teacher traces across problems?**
>
> It corresponds to distillation from only the best single teacher, and we are clarifying this in the new version. Also note that for each problem, we have already used the ground-truth final answer to only keep the correct trajectories. Given this, selecting “best teacher per problem” is a very hard combinatorial optimization problem since we need to choose from {teacher 1, teacher 2} for each problem such that when trained on all problems the performance is the best. Let us know if you meant something else!
>
> We hope these clarifications and new results help address your main concerns about clarity and RL performance and make it easier to assess the core contributions of our work. In summary, we (1) formulate mode collapse in parallel test-time scaling as a compute allocation problem, (2) propose mode-conditioning (ModC) as a way to explicitly balance test-time compute across diverse reasoning modes (via separate models or mode prefixes), and (3) introduce gradient-based mode discovery as a practical way to uncover such modes even when they are not labeled. Empirically, we show that ModC consistently improves pass@k under a fixed compute budget on both synthetic and real math benchmarks, and the gain holds after RL (including both standard RL and pass@k RL).
>
> Thanks for being willing to raise your score, we appreciate it! We hope we have addressed your concerns. Please let us know if you have any further questions!

---

### Official Review · Reviewer_h3Wb · 2025-11-05

**Soundness:** 3
**Presentation:** 2
**Contribution:** 2
**Rating:** 2
**Confidence:** 3

**Summary:**

The paper presents a simple, yet effective approach to diversify the model’s outputs by providing explicit control over the modes. It explores two methods: 1) training separate specialist models and splitting test-time compute between them, 2) training a single model with mode-specific prefixes, and sample equally with the corresponding prefixes. The paper shows that these approaches surpass the mixed model trained with both modes. Morover, in the case of unspecified modes, the paper proposes an automatic mode discovery method based on gradient clustering. It shows that the method captures the labels reliably and further mode-conditioning on them recovers the improvements.

**Strengths:**

The paper pinpoints a simple but important suboptimality in training language models with diverse data. It verifies the intuition with experiments with both toy settings such as different strategies for Countdown, and with real-world tasks and traces distilled from teachers. It is comprehensive in experimenting with different forms of chain-of-thought (short and long) generated with different models. Moreover, the work pushes its practical relavance further by providing a method for discovering unobserved modes in the data based on gradient clustering, which makes the idea more generalizable to different settings.

**Weaknesses:**

The paper could improve its presentation by defining its metrics more clearly. For example, it’s not clear how the “Fraction of BFS per problem” metric is computed for Figure 2. In section 5.1, p_\theta is not defined, so it’s not obvious how the gradient is computed.

I also did not understasnd how heuristic prunings and search budget constraints make the problems solvable with only one of BFS and DFS, making it unclear why this setting captures the diverse setting desired.

The novelty of the idea to learn separate models and aggregating them instead of learning from a mixed dataset is questionable given the literature around mixture of experts and other works such as “Mix Data or Merge Models? Optimizing for Diverse Multi-Task Learning”.

**Questions:**

1. Could you please explain how the heuristic prunings limit the solution to one of BFS and DFS?
2. How is the ‘fraction of BFS used’ computed?
3. For the distilling experiments, what kind of prefix do you use for different teachers? How does knowledge sharing happen in those 4. experiments if the model learns to follow one strategy given a prefix?
4. Could you explain how the gradient is computed in the gradient clustering method?
5. Did you run the gradient clustering method for the long-CoT datasets too?

---

> ### Author Response · Authors · 2025-11-22
> **Author rebuttal (part 1)**
>
> Thank you for your thoughtful review. We appreciate your recognition of the important suboptimality we identify, the comprehensive nature of our experiments, and the practical relevance of our gradient-clustering approach. We also deeply appreciate your feedback on what parts of the presentation were unclear. We have expanded below on all the points you raised and have also updated the paper to make the presentation clear. We hope the updates help the contribution come through more clearly and lead to a more favorable view of the work.
>
> > **Could you please explain how the heuristic prunings limit the solution to one of BFS and DFS?**
>
> We used an algorithm that runs BFS/DFS but uses heuristics to prune. At each expansion step, only nodes with the lowest heuristic scores are kept. Since the heuristics does not look at all possibilities globally, the true solution has a chance to be pruned. In such cases, the search algorithm would never reach the target. Here we provide a concrete example: one of the heuristics used by the search algorithm is called “sum heuristics”, which compares the sum of remaining numbers with the target. Let’s say we start with numbers {10, 10, 4, 4} to compute target 24. The only correct solution is (10 * 10 - 4) / 4 = 24. When exploring the first step, BFS would try all possible pairs and one operation (+, -, *, /), and only keep those whose sum is closer to 24. For example, when it tries 10 + 4 as the first step, then the set becomes {14, 10, 4}, whose sum is 28, so the heuristics score is |28 - 24| = 4. It is easy to show that the worst trial is 10 * 10 under this sum heuristics since it gives {100, 4, 4}, whose sum is 108, so the heuristics score is |108 - 24| = 84 which is the largest; so 10 * 10 is pruned in the first step. However, 10 * 10 is actually the first operation in the only true solution, making BFS + sum heuristics unable to solve the problem. We are adding this clarification in the new version.
>
> > **How is the “fraction of BFS used” computed?**
>
> We apologize for the lack of clarity. This is an important experiment to show the imbalance of test-time compute budget allocation for each mode, and show that ModC can largely mitigate this. We are revising the paper to clarify this. For each test problem we sample from the model 256 times, and detect for each sample whether the model used DFS or BFS. To do so, we reconstruct the search trajectory from the model’s generation. If the second visited node is one level below the first visited node, then it’s DFS; if the second visited node is at the same level as the first visited node, then it’s BFS; otherwise, it’s invalid (which we found to happen 3% of time). This gives us the fraction of BFS samples for that problem, and then we plot the distribution of that fraction over all problems. Moreover, sampling 256 times ensures that the estimation of fraction has a very low variance.
>
> > **For the distilling experiments, what kind of prefix do you use for different teachers? How does knowledge sharing happen in those experiments if the model learns to follow one strategy given a prefix?**
>
> In the teacher-distillation experiments, each teacher is associated with a short prefix, “using {teacher-name}-style reasoning\n”. Knowledge sharing happens because the weights are shared for the two modes (only the prefix changes), and we expect knowledge sharing to help capabilities irrelevant to the strategy/style, e.g., arithmetics, simplification of math notations.
>
> > **Could you explain how the gradient is computed in the gradient clustering method?**
>
> The gradients are computed wrt the base mode before finetuning. Specifically, we feed both the input and the output to the base model, compute the cross-entropy loss only on the output tokens, and then do backprop. We then flatten the gradient and reduce the dimension (to maintain a reasonable disk storage) using inner-product-preserving projection. Clustering is done on the projected gradients.
>
> > **Did you run the gradient clustering method for the long-CoT datasets too?**
>
> We agree this is an important extension and appreciate the suggestion. Practically, we face compute constraints for long-CoT gradient clustering: only the 7B model gives us meaningful numbers for long-CoT, but gradient projection for the 7B models is not possible with the GPUs we have. FYI, for short-CoT we use the 1.5B model as proxy for the 7B model, but this does not transfer well in the long-CoT setting.

---

> ### Author Response · Authors · 2025-11-22
> **Author rebuttal (part 2)**
>
> > **The novelty of learning separate models and aggregating them instead of learning from a mixed dataset is questionable given the literature on mixture-of-experts and works such as “Mix Data or Merge Models? Optimizing for Diverse Multi-Task Learning”.**
>
> We would like to point out that our key insight is that ignoring multiple modes is a big blindspot in current reasoning work. One of our compelling examples is how we show that in the standard setup, people believe we can just use the best teacher and it works better, but we show that it's in fact not true - we are able to use diverse teachers and improve inference-time scaling. The best-performing variant of our method on math does not train separate models, but conditions the model on different modes. Also, it’s unclear if mixture-of-experts/model merging can improve diversity and make the model using different modes in a more balanced way at test time. That said, we agree that mixture-of-experts works are conceptually relevant and will add discussion.
>
> We hope these clarifications help address your main concerns about clarity and make it easier to assess the core contributions of our work. In summary, we (1) formulate mode collapse in parallel test-time scaling as a compute allocation problem, (2) propose mode-conditioning (ModC) as a way to explicitly balance test-time compute across diverse reasoning modes (via separate models or mode prefixes), and (3) introduce gradient-based mode discovery as a practical way to uncover such modes even when they are not labeled. Empirically, we show that ModC consistently improves pass@k under a fixed compute budget on both synthetic and real math benchmarks, and the gain holds after RL (including both standard RL and pass@k RL).
>
> We would like to share a new experiment that we find exciting! Beyond the original SFT/distillation experiments, we now show that in the 0.5B math setting ModC preserves its advantages even after RL (both standard RL and pass@k RL training; see responses to Reviewers MR1M and ghYh), where we demonstrate that:
>
> 1. Crucially, while pass@k RL brings both models to the same pass@1, ModC excels immediately at k=2 with higher pass@k scores. This shows that unlike standard SFT, ModC successfully enriches the solution space without degrading the accuracy of its top output.
> 2. The finding also holds for standard RL.
>
> Let us know if you have other questions!

---

### Official Review · Reviewer_MR1M · 2025-11-06

**Soundness:** 3
**Presentation:** 3
**Contribution:** 3
**Rating:** 6
**Confidence:** 4

**Summary:**

The paper proposes mode-conditioning, a test-time inference strategy that allocates a certain number of samples to each mode in order to improve diversity of samples, mitigating the issue of mode collapse. The authors show that ModC leads to consistent gains across tasks both when modes are fixed, as well as being able to discover modes automatically via gradient clustering.

**Strengths:**

- The ModC method is novel, creative, and effective, addressing the critical issue of lack of diversity.
- The authors demonstrate that ModC training works well on a variety of tasks. They explore the idea throughout a variety of settings and domains, and it performs above standard baselines in all cases. The technique seems to be quite general and could have potential downstream applications beyond those listed in the paper.
- The authors also compare different ways of implementing mode conditioning, and do a thorough analysis on other factors like model size, CoT length, etc.

**Weaknesses:**

- The work does not have any comparisons with other diversity-inducing techniques, for example pass@k training (https://arxiv.org/abs/2508.10751) or optimal sample allocation (https://arxiv.org/abs/2410.22480). While ModC is evidently effective against simple baselines, it is difficult to understand the advantages and disadvantages of this method against some of these other methods.

**Questions:**

- Most of the ablations are comparing variants of ModC. Could you provide a comparison of ModC against other diversity-inducing techniques (see comment in weaknesses section)?
- Do you see a clear diversity increase after ModC? For example, for MATH500, if you consider how many distinct answers are produced for each problem, how much does it increase with ModC?
- Does the idea also apply to other domains, such as code generation?

---

> ### Author Response · Authors · 2025-11-22
>
> Thank you for your thoughtful review. Our submission showed that mode-conditioning (modC) during SFT improves over SFT on a variety of reasoning datasets. We are excited to share new results demonstrating that modC also provides a markedly better starting point for RL, enhancing both standard fine-tuning and diversity-preserving approaches.
>
>
> > **The work does not have any comparisons with other diversity-inducing techniques.**
>
> We appreciate this important point and added experiments over the past week that we hope addresses this concern. First, we would like to clarify that our proposed ModC is an SFT method and we are not aware of other diversity-inducing techniques at the SFT stage. Most diversity-inducing interventions involve changing the RL process. This includes pass@k training that you mention but also some other recent ones such as inference-aware finetuning (Chow et al. 2025). Conceptually, these interventions are designed to prevent diversity collapse during RL and perhaps cannot recover modes that are missed in the SFT starting point itself.
>
> A better SFT starting point via ModC should translate to gains after RL as well. We ran new experiments this week to verify if ModC’s benefit holds after pass@k training (or even after standard RL, which we also verified). We updated the results in Figure 7 of the latest version, and summarize the findings below:
>
> 1. Crucially, while pass@k RL brings both models to the same pass@1, ModC excels immediately at k=2 with higher pass@k scores. This shows that unlike standard SFT, ModC successfully enriches the solution space without degrading the accuracy of its top output.
> 2. The finding also holds for standard RL.
>
> There are some other methods like OSCA that you linked which is a diversity-inducing technique at inference time. This is again orthogonal to ModC, which is an SFT intervention. We expect ModC to show gains when combined with OSCA. We prioritized the RL experiments over the past week since those are more general and widely impactful/interesting, but we are currently working on integrating with OSCA and other inference-time approaches as well, and would add these to our final version.
>
> We hope you are also excited by our RL results and are convinced that mode-conditioning is a general, scalable and powerful way to further boost the gains provided by diversity-inducing techniques.
>
> > **Do you see a clear diversity increase after ModC? For example, for MATH500, if you consider how many distinct answers are produced for each problem, how much does it increase with ModC?**
>
> We argue that the final answer itself is often unique as for math problems there is a single correct answer to the question. However, we can test whether the LLM employs diverse reasoning traces to arrive at the answer. There is no clean way to do this but we show a case study: for the problem “Convert the point $(0,3)$ in rectangular coordinates to polar coordinates.”, the model trained with standard SFT on mixed teacher data begins with restating the information “The point …” 86% of time, while the model trained with ModC begins with the restatement only 45% of time and with recalling how to convert rectangular coordinates to polar coordinates (e.g., “To convert …, we need to…) 42% of time.
>
> > **Does the idea also apply to other domains, such as code generation?**
>
> Yes, the method is not specific to math, as it only assumes there exist different modes of solving a task. For example, in coding, we can expect one mode to prefer the imperative programming paradigm while the other to prefer the functional paradigm; for proof generation, proof by construction and proof by contradiction are two distinct modes. We expect the method can be extended to code and proof generation.
>
> We hope the additional experiments on pass@k RL and standard RL address most of your concerns. Please let us know if you have any further questions!

---

### Author Response · Authors · 2025-12-03
**Message to AC**

Dear Area Chair,

We greatly appreciate the reviewers' thoughtful feedback and are excited to share that we have substantially strengthened our paper during the rebuttal. While we unfortunately did not hear back from the reviewers after our responses, we want to highlight how we believe we have comprehensively addressed their concerns.

**Key experimental improvements:**

We proposed mode conditioning (ModC), an SFT and test-time method that discovers and teaches the model to internalize different modes in the data to improve test-time compute allocation. In the latest version, we conducted new RL experiments (the reviewers' primary concern) demonstrating that ModC's benefits persist after both standard RL and diversity-inducing, pass@k RL training. Crucially, after both RL and pass@k RL training, ModC excels the baseline immediately at k≥2, and the gap increases with k. Moreover, the pass@k gain does not come at a cost to pass@1 unlike other diversity methods such as increasing temperature. It shows that ModC successfully enriches the solution space without degrading top-output quality. These results are now in updated Figure 7.

**Addressing reviewers' concerns:**

**R1 (score: 6)** The reviewer’s main concern was about lacking comparisons with diversity-inducing baselines. While there are no canonical baselines in the SFT stage, our new RL experiments (see **key experimental improvements**) demonstrate that ModC provides a superior starting point that amplifies gains from diversity-preserving RL methods like pass@k training that the reviewer kindly suggested as a baseline to consider.

**R2 (score: 2)** had a low score but concerns were primarily about clarity in presentation; they didn’t request additional experiments. We substantially revised all technical explanations they flagged to have more details: how heuristic pruning works, how BFS fractions are computed, implementation details of the prefix version of our method, and gradient clustering details.


**R3 (score: 4; indicated willingness to raise score)** explicitly stated they would raise their score if we address their concerns. Their concerns were around clarity which we have substantially rewritten / added technical content - all the interpretations of the reviewer are correct, and our comparisons are also compute matched. We also directly addressed their concern about RL: the reviewer said they’d be happy with just the synthetic countdown task, but we were able to run experiments on real math post-training settings and evaluate on MATH500 (see **key experimental improvements**). We find that ModC can boost pass@k of both standard RL and diversity boosting RL, without sacrificing pass@1 accuracy unlike other diversity methods such as increasing temperature. We have not left any concern or point unaddressed, and we believe the reviewer would have been especially excited about RL experiments

**R4 (score: 8)** had minor concerns about AIME evaluation and connections to prior work, which we addressed by adding relevant discussions.

**Core contribution of our work:** We identify a fundamental blindspot in current reasoning work—mode collapse during test-time scaling—and propose a simple, general solution that **works across SFT, distillation, and RL settings**. Our gradient-based mode discovery makes this practical even without labeled modes.

We believe our rebuttal directly addresses all major concerns, and also satisfies the explicit conditions reviewers stated for score increases. Thank you for your consideration!

---

### Meta-Review · Area_Chair_5cXn · 2026-01-07

**Summary:**

The paper addresses the "diversity collapse" problem in test-time compute scaling, where simply increasing sample counts yields diminishing returns due to mode repetition. The authors propose "Mode-conditioning" (ModC), a framework that explicitly allocates compute to different reasoning modes (via specialist models or prefixes). Crucially, they introduce a gradient-based clustering method to discover these modes without ground-truth labels.
The original submission demonstrated effectiveness in SFT settings. During the rebuttal, the authors added significant experiments showing that ModC serves as a superior starting point for Reinforcement Learning (RL), outperforming baselines when combined with diversity-inducing RL methods (e.g., pass@k training).

**Reviewer Concerns:**

Concerns Addressed by Rebuttal:

1. Lack of RL / Post-training evaluation (R1, R3): This was the most critical concern. R1 and R3 questioned whether the diversity gains in SFT would survive the distribution collapse typical of RL. The authors included new experiments (Figure 7 in the revision) showing that ModC combined with pass@k RL significantly outperforms baselines at k≥2, effectively addressing this concern.
2. Comparison with other diversity-inducing techniques (R1): The authors clarified that ModC is an SFT-stage intervention orthogonal to inference-time methods. The new data shows ModC amplifies the benefits of methods like pass@k training, rather than just competing with them.
3. Compute-matching and Fairness (R3): The authors clarified that the "specialist models" approach uses the same total training FLOPs (steps are halved per model) as the dense baseline, addressing the fairness concern.
4. Clarity of Gradient Clustering & Heuristics (R2): The authors provided detailed explanations of the gradient computation and the heuristic pruning logic in the revision, improving the presentation significantly.

Outstanding Concerns:
1. Complexity vs. Gain (Implicit in R2): While effective, the method requires either training multiple models or a complex pre-processing step (gradient clustering) to find modes. Some practitioners might still prefer simpler inference-time diversity heuristics (like elevated temperature or diversity penalties) which don't require changing the training pipeline, although the paper argues ModC provides a better Pareto frontier.
2. Baselines: While pass@k RL was added, broader comparisons to other advanced inference-time search algorithms (beyond standard sampling) could still strengthen the paper, though this may be out of scope for a single submission.

**Reviewer Scores:**

- h3Wb (2->4). This reviewer was the most negative, citing clarity and novelty. The clarity issues were fixed. They might still hold reservations about the "novelty" against MoE literature, but the specific application to test-time scaling distinct from standard MoE routing justifies a higher score than rejection.
- ghYh (4->6). This reviewer explicitly stated: "I am willing to improve my score if the authors can meaningfully address [RL post-training]." The authors did exactly this with the new Figure 7 results.

There is no clear evidence that the other reviewers will update the score.

---

### Decision · Program_Chairs · 2026-01-26

Accept (Poster)